# Effect of Enzymatic Treatment of *Chrysanthemum Indicum* Linné Extracts on Lipid Accumulation and Adipogenesis in High-Fat-Diet-Induced Obese Male Mice

**DOI:** 10.3390/nu11020269

**Published:** 2019-01-25

**Authors:** Ji-Hyun Lee, Joo-Myung Moon, Yoon-Hee Kim, Bori Lee, Sang-Yong Choi, Bong-Joon Song, Dae-Ki Kim, Young-Mi Lee

**Affiliations:** 1Department of Immunology and Institute for Medical Sciences, Chonbuk National University Medical School, Jeonju 54907, Korea; jihyunsh1211@naver.com; 2BTC Corperation, #703, Technology Development Center, 705 Haean-ro, Sangnok-gu, Andsan-si 15588, Korea; mhjj1919@btcbio.com (J.-M.M.); kyh@btcbio.com (Y.-H.K.); 3Department of Oriental Pharmacy, College of Pharmacy and Wonkwang-Oriental Medicines Research Institute, Wonkwang University, Iksan 54538, Korea; leebori1004@naver.com; 4Wonkwang Pharmaceutical Co., Ltd., Iksan 54588, Korea; choisy0115@naver.com; 5Department of Food Science and Biotechnology, Wonkwang University, Iksan 54538, Korea; twinf1@hanmail.net

**Keywords:** *Chrysanthemum Indicum* Linné, obesity, enzymatic treatment, adipogenesis, lipid accumulation

## Abstract

Enzyme treatment of the foods and herbs has been used to improve the absorption rate the efficiency of plant extracts by converting the glycosides of the plant into aglycones. In this study, we examined the obesity-inhibitory effect of *Chrysanthemum indicum* Linné (CI) treated with enzymes such as viscozyme and tannase, which are highly efficient in converting glycosides to aglycones and then compared with untreated CI extract. The enzyme-treated CI ethanol extract (CIVT) was administered orally at various doses for 7 weeks in the high fat diet (HFD)-fed male mice. CIVT administration reduced the body weights, the food efficiency and the serum levels of lipid metabolism-related biomarkers, such as triglycerides (TG), total cholesterol (TC), low-density lipoprotein cholesterol (LDL-c) and leptin in the dose-dependent manner but not those high-density lipoprotein cholesterol (HDL-c) and adiponectin. CIVT also reduced considerably the total lipid amount in the liver and the size of adipocytes in the epididymal white adipose tissue (eWAT). CIVT effectively downregulated the adipogenesis-related transcription factors such as peroxisome proliferation activated receptor (PPAR)-γ and CCAAT/enhancer binding protein-α (C/EBP-α) but up-regulated PPAR-α, in the liver and eWAT. In addition, when compared to the enzyme-untreated CI 50% ethanol extract (CIEE), CIVT enhanced the reduction of body weight and lipid accumulation. Moreover, the viscozyme and tannase treatment of CI increased the flavonoid contents of the aglycone form. Therefore, our results support that the enzymatic treatment induced the production of aglycones for potentially suppressing the adipogenesis and lipid accumulation in HFD-fed mice. It suggests that CIVT might be an effective candidate for attenuating the over-weight and its related diseases.

## 1. Introduction

In plants and plant-derived foods, flavonoids are predominantly in the form of glycosides. But, glycosides have a disadvantage of low bioavailability because of their high polarity and poor absorption ability in human body. These glycosides should be transformed into aglycones (fermented glycoside) to increase bioavailability. Recent studies have reported that treating plants with enzymes increases the contents of various bioactive ingredients [1]. Several studies have selected enzymes such as viscozyme and tannase to increase the bioactive ingredients of various plants [1,2,3]. Viscozyme is a multiple enzyme complex containing a variety of carbohydrate hydrolases [2] and tannase serves to break the ester linkage between the various compounds [3].

Obesity is a state of energy imbalance within the body that occurs as a result of over- ingestion or when energy expenditure is less than the energy ingested. This leads to an abnormal accumulation of body fat associated with expansion of adipose tissue [4]. The causes of obesity are related to hormone changes, genetic factors, environmental factors and mental health problems and obesity has recently been linked to important health problems causing metabolic diseases such as hypertension, type 2 diabetes, hyperlipidaemia, atherosclerosis and other chronic diseases [5,6,7].

Adipogenesis and intracellular lipid accumulation regulate lipid metabolism and are also involved in the development of obesity [8]. The increase in the size of adipose tissue is caused by the processes of enlargement of adipocytes through lipid accumulation (hypertrophy) and increase in the number of mature adipocytes by proliferation and differentiation of undifferentiated adipocytes (hyperplasia) [9]. Adipogenesis is a differentiation process that converts pre-adipocytes (mesenchymal precursor cells) into mature adipocytes, accompanied by adipocyte morphology changes and lipid accumulation [10]. To differentiate pre-adipocytes into adipocytes, transcription factors are needed; these transcription factors include PPAR-(peroxisome proliferation activated receptor-) and C/EBP-(CCAAT / enhancer binding protein-) [11,12]. In addition, increased expression of transcription factors activates adipocyte specific genes, such as leptin and adiponectin, to regulate adipogenesis [13]. The liver is one of important organs to regulate adipogenesis. It plays a crucial role in lipid metabolism and accumulation through the lipid synthesis and fatty acid oxidation [14]. Many transcription factors such as PPAR-α, PPAR-γ and C/EBP-α are also involved in the initiation of adipogenic differentiation [15].

*Chrysanthemum indicum* flower (CI) is an herb used widely in East Asia to treat various diseases. Studies have suggested that CI has many functions including anti-viral, anti-oxidant and anti-inflammatory activities [16,17]. Recently, it was reported that an ethanol extract and ethyl acetate fraction of CI have anti-obesity effects in a mouse model of high fat diet (HFD)-induced obesity [18,19]. However, these studies have limitations for the clinical trials due to a large amount of CI extract. Thus, the enzymatic treatment is considerable to solve these limitations in its bioavailability.

In this study, we tried the enzymatic treatment with the cell wall digestion enzyme viscozyme and tannin dehydratase tannase to convert glycosides to aglycones in CI dried powder. Anti-obesity ability of CI ethanol extract after enzymatic treatment (CIVT) was examined in a mouse model of HFD-induced obesity, In addition, we analysed the contents of CIVT components and compared them of natural CI ethanol extract (CIEE).

## 2. Materials and Methods

### 2.1. Materials and Reagents

The primary antibodies, mouse anti- PPAR-α (#sc-398394), PPAR-γ (#sc-J3012), C/EBP-α (#sc-65318), and, β-actin (#sc-47778), were purchased from Santa Cruz Biotechnology (Santa Cruz, CA, USA) and used at 1:1000 dilutions. The secondary antibody for mouse primary antibodies, goat-anti mouse IgG F(ab)2’ HRP conjugate antibody, was purchased from Enzo Life Sciences (Enzo Life Sciences, Farmingdale, NY, USA). Secondary antibody was used at 1:5000 dilutions. Most of chemicals were purchased from Millipore sigma (St. Louis, MO, USA).

### 2.2. Preparation of the Ethanol Extract of Enzymatic CI (CIVT)

Four litres of purified distilled water was added to 200 g of dried CI powder and boiled for 2 h at 90 °C in water bath. After cooling to 45 °C, it was adjusted to pH 4.5 using citric acid solution (1N). Then, 3 g of viscozyme (Viscozyme-L. Novozyme, Denmark) and 400 mg of tannase (Tannase-KTFH, Kikkoman, Japan) were added and the enzyme treatment was carried out with shaking at 100 rpm for 24 h at 45 °C. After the enzyme treatment was completed, the enzyme was inactivated for 30 min at 90 °C and the primary extract was prepared by filtration using a standard sieve (No. 850, 75 μm). Four litres of 50% ethanol were added to the remaining residual substance and the mixture was extracted for 6 h at 70 °C. Then, the substance was mixed with the primary extract, concentrated under reduced pressure, sterilized for 15 min at 121 °C and lyophilized to prepare an enzymatic CI extract (CIVT). Through three repeated experiments, the yield and components were analysed to standardize the CIVT. CI 50% ethanol extract (CIEE) was prepared as previously described [15]. The dried powders of CIVT and CIEE were stored at 4 °C until use. CIVT was provided from BTC corporation (Sangnok-gu, Ansan 15588, Korea).

### 2.3. High Performance Liquid Chromatography (HPLC) Analysis

To perform the HPLC analysis, the Elite Lachrom HPLC-DAD system equipped with UV (DAD) detector (Hitachi High-Technologies Co., Tokyo, Japan) was used. The analytical column used was X-selects HSS C18 (4.6 × 250 mm, 5 μm, Waters, Milford, MA, USA). The column temperature was at 30 °C and the injection volume was 10 μl. The mobile phases were composed of solvent (A) = 0.1% Phosphoric acid in distilled water and solvent (B) = Acetonitrile. The run time was 45 min and the mobile phase process gradient flow was as follows: (A)/(B) = 85/15 (0 min) → (A)/(B) = 60/40 (0–25 min) → (A)/(B) = 0/100 (25–30 min) → (A)/(B) = 85/15 (30–45 min). The mobile phase flow rate was 1.0 ml/min and the wavelength of the UV (DAD) detector was set at 330 nm.

### 2.4. Animals and Treatment

Six-week-old, male C57BL/6J mice (22 ± 2 g) were purchased from SAMTAKO Bio Korea (Osan, South Korea). Mice were housed in a controlled environment with constant humidity (50 ± 10%) and constant temperature (22 ± 2 °C), under a 12-h dark/12-h light cycle, with a standard laboratory diet and water supply. After a one-week acclimation period, Mice were randomly divided into seven groups of six mice each. All mice were housed in groups and all treatment groups were vehicle-controlled. (1) ND, mice fed with normal chow diet; (2) HFD, mice fed with high-fat diet (HFD; 45% lard oil in normal chow diet); (3) CIVT-4, mice fed with HFD and treated with CIVT (4 mg/kg); (4) CIVT-20, mice fed with HFD and treated with CIVT (20 mg/kg); (5) CIVT-100, mice fed with HFD and treated with CIVT (100 mg/kg); (6) CIEE-100, mice fed with HFD and treated with CIEE (100 mg/kg); (7) ORL, mice fed with HFD and treated with orlistat (20 mg/kg); Orlistat is an effective drug for obesity that inhibits lipid accumulation [20]. Since many studies have reported that orlistat inhibited obesity, we selected orlistat as a positive control during the in vivo experiment [21]. We compared the difference between CIVT and CIEE at each 100 mg/kg dose. Each sample was dissolved in 0.3% carboxymethylcellulose (CMC) and orally administered once a day for seven weeks. To measure food intake, we measured every week the weight of food that the mice consumed for a week. Body weight food intake and food efficiency ratio were measured once a week over the seven-week experimental period. (Food efficiency ratio = Gained body weight (g)/Food intake during trial period (g)) The animal experimental protocol was approved by the Institutional Animal Care Committee of Chonbuk National University (CBNU-IACUC) (Approval No. CBNU 2015-0010).

### 2.5. Micro Computed Tomography (Micro-CT)

Before being sacrificed, all mice were fasted for 8 h and anesthetized by intraperitoneal injection (i.p.) of ketamine (75 mg/kg) and rompun (15 mg/kg). In vivo micro-CT images were acquired using a Skyscan1076 micro-CT scanner (Skyscan, Aartselaar, Belgium). CT was performed using the following conditions: 35 μm pixel size; 50 kVp source voltage; 200 μA source current. The X-ray detector composed a 12-bit water-cooled charge-coupled device (CCD) with a scintillator and high-resolution camera (4000 × 2300-pixel). All images were obtained with increments of 0.6 degrees and exposure time of 0.46 sec using a 1-mm aluminium energy filter.

### 2.6. Serum and Tissue Preparation

After micro-CT scanning, the mice were anesthetized by exposure to diethyl-ether and euthanized by rapid cervical dislocation. Blood samples were taken from the inferior vena cava of the mice. The collected blood samples were held for 1 h at room temperature. Then, serum was obtained by centrifugation at 3000 rpm for 10 min at 4 °C. After death was confirmed, organs (liver, kidney and spleen) and abdominal fat were removed and washed with cold saline solution. Isolated fat and organs were measured using a weighing beam. After all fat tissues were divided into four equal parts, one piece of fat tissue was fixed in 4% formaldehyde solution at room temperature for H&E staining. Serum and remaining organ samples were stored at −80 °C until use.

### 2.7. Serum Biochemical Analysis

Serum biochemical levels of triacylglycerol (TG), total cholesterol (TC) and high-density lipoprotein-cholesterol (HDL-c) were analysed using commercial detection kits (Asan Pharmaceutical Co., Seoul, South Korea). All of measurements were performed according to the manufacturer’s instructions. Low-density lipoprotein cholesterol (LDL-c) level was calculated using Friedewald’s formula [22]; LDL-c (mg/dl) = TC-(TG/5)-HDL-c, The total lipid levels in liver tissue were measured according to Folch’s method [23]. Total lipids were extracted with a mixture of chloroform: methanol (2:1, *v*/*v*) and individual total lipids were quantified by gravimetric analysis.

### 2.8. Enzyme-Linked Immunosorbent Assay (ELISA)

Serum levels of leptin and adiponectin were analysed using leptin ELISA kit (R&D Systems, Inc., Minneapolis, MN, USA) and adiponectin ELISA kit (R&D Systems, Inc., Minneapolis, MN, USA), according to the manufacturer’s protocols.

### 2.9. Histological Analysis

It is observed that the eWAT is markedly increased in abdominal cavity of obese-induced mice. Therefore, white adipose tissues of all mice were analysed by observing representatively eWAT. eWAT specimens were fixed in 4% formaldehyde solution at room temperature overnight, embedded in paraffin wax. Then, the tissues were sectioned serially at 10 μm and stained with haematoxylin and eosin (H&E; haematoxylin, 4 min and eosin, 2 min) for observation of histological alterations. Images were observed at ×100 magnification and photographed on an Olympus CX21 microscope (Olympus America Inc., Melville, NY, USA). The adipocyte sizes from epididymal white adipose tissue were quantitated using Adobe Photoshop CS5 (Adobe Systems Inc., CA, USA).

### 2.10. Western Blot Analysis

Liver and eWAT tissues were homogenized and lysed in ice-cold PRO-PREP lysis buffer (iNtRON Biotechnology, Seongnam, Korea), containing phosphatase inhibitors and a protease inhibitor cocktail for 30 min at 4 °C. Then, the lysates were centrifuged at 12,000 rpm for 20 min at 4 °C and the supernatants of these tissue lysates were separated. The tissue protein quantification was measured using Bicinchoninic Acid Protein Assay Kit (Sigma-Aldrich, St. Louis, MO, USA). Each 20 μg of protein samples were separated by 10% sodium dodecyl sulphate-polyacrylamide gel electrophoresis (SDS-PAGE) and transferred to polyvinylidene difluoride (PVDF) membranes (GE Healthcare, Chicago, IL, USA). The membranes were blocked using 5% bovine serum albumin (BSA) in tris-buffered saline containing 0.1% Tween-20 (TBST) for 1 h at room temperature and incubated with primary antibodies against PPAR-α, PPAR-γ, C/EBP-α and β-actin overnight at 4 °C. Afterward, the membranes were washed two times with TBST, incubated with horseradish peroxidase (HRP)-conjugated secondary antibody for 1 h at room temperature and rewashed four times with TBST and one time with TBS. The proteins were visualized using an enhanced chemiluminescence (ECL) detection kit (BIO-RAD, Hercules, CA, USA) and a Davinch-*In vivo* & western imaging system (Davinch-K, Seoul, South Korea).

### 2.11. Statistical Analysis

All data are presented as means ± S.E.M. (standard error of the mean) and were analysed using Graph Pad Prism software 5.0 (Graph Pad Software, Inc., La Jolla, CA, USA). Statistical significance was tested using one-way analysis of variance (ANOVA) with Bonferroni correction for *post-hoc* analysis to determine differences between groups. *p* < 0.05 was considered to indicate statistically significant differences (* *p* < 0.05, ** *p* < 0.01 and *** *p* < 0.001).

## 3. Results

### 3.1. CIVT Reduces Body Weight Gain and Food Efficiency Ratio in HFD-Fed Obese Mice

To examine the anti-obesity effect of CIVT, body weight and food intake were measured weekly. On the first day of the experiment (week 0), the average body weights of all groups were similar. The final average body weights of all groups had increased from the start day. In particular, body weight in the HFD group was far higher than that in any other group (Figure 1A). At week 7, the mean body weight gain value was increased by ND (6.95 ± 0.46), HFD (14.20 ± 0.71), CIVT-4 (10.9 ± 0.77), CIVT-20 (6.58 ± 0.48), CIVT-100 (6.35 ± 0.57) and orlistat (ORL) (7.15 ± 0.22), respectively (Figure 1B). CIVT oral administration considerably suppressed weight gain, in a dose-dependent manner. As shown in Figure 1C, no marked food intake differences were observed between the HFD group and CIVT oral-administered groups. CIVT orally administered groups showed dose-dependent suppression of food efficiency ratio, compared with the HFD group (Figure 1D). These results suggest that CIVT did not reduce food intake but reduced the food efficiency ratio.

### 3.2. Effects of CIVT on Organ Weight in HFD-Fed Obese Mice

After the mice were sacrificed, the weights of their adipose tissue and organs were measured. As shown in Table 1, HFD induced a weight increase in all organs (eWAT, liver, spleen and kidney). CIVT decreased eWAT, liver, spleen and kidney weight in HFD-fed obese mice, in a dose-dependent manner. In particular, CIVT (20 and 100 mg/kg) oral administration reduced organ weight.

### 3.3. CIVT Regulates Serum Lipid Parameters in HFD-Fed Obese Mice

Among obese persons, a higher degree of obesity often results in lipid metabolism abnormalities such as elevated serum TG, TC, LDL-c and leptin levels and decreased HDL-c and adiponectin levels [24]. So, to evaluate the effect of CIVT on HFD mice, serum lipid metabolism-related biochemical parameters were measured. As shown in Figure 2, compared to the ND group, the HFD group showed considerably increased serum levels of TG, TC, LDL-c and leptin and decreased serum levels of HDL-c and adiponectin. CIVT oral administration inhibited TG, TC and LDL-C levels and increased HDL-c levels (Figure 2A–D), compared to levels in the HFD group, in a dose-dependent manner. In addition, CIVT dose-dependently decreased leptin levels and increased adiponectin levels as compared with levels in the HFD group (Figure 2E,F).

### 3.4. CIVT Decreases Fat Deposition, Adipocytes Size and Total Lipid Levels in the Liver Tissues in HFD-Fed Obese Mice

To investigate the effect of CIVT on fat deposition, micro-CT was used. As shown in Figure 3A, the amount of visceral adipose tissue was dramatically increased in the HFD group, as compared to that in the ND group. However, CIVT oral administration dose-dependently suppressed fat volume. As a result of measuring the volume using the micro-CT program, it was confirmed that the orbital fat volume (%) was dose-dependently and sharply decreased by oral administration of CIVT, corroborating the previous result (Figure 3B). Histological alterations of eWAT were observed using H&E staining and the size of adipocytes size from each group was measured using photoshop program (Figure 3C,D). The average size of adipocytes in the HFD group was enlarged compared with that in the ND group. In contrast, the CIVT orally administered groups showed a decrease in the size of their adipocytes, in a dose-dependent manner. These results suggest that CIVT inhibits adipogenesis and reduces the size of adipocytes. To examine the inhibitory effect of CIVT on the increase in liver fat accumulation in HFD-fed mice, total lipid levels in liver tissues were analysed. As shown in Figure 3E, the total lipid levels in the liver tissues were increased in the HFD group compared with those in the ND group. However, the CIVT group (4, 20 and 100 mg/kg) attenuated the total lipid levels of the liver tissues in a dose-dependent manner. In particular, the CIVT-100 was more effective than the positive control ORL. This suggests that CIVT inhibits lipid accumulation in liver tissue.

### 3.5. CIVT Regulates the Levels of Adipogenesis-Related Proteins in Liver Tissues and in eWAT, of HFD-Fed Obese Mice

To examine the anti-obesity mechanisms of CIVT in liver tissues and eWAT, we measured adipogenesis-related proteins such as PPAR-α, PPAR-γ and C/EBP-α, using western blotting. As shown in Figure 4, the protein expression level of PPAR-α in liver tissues and eWAT was lower in the HFD group than in the ND group. However, PPAR-α level was increased in the CIVT orally administered group and ORL group compared to that in the HFD group. In addition, the protein expression levels of PPAR-γ and C/EBP-α were increased in the HFD group, compared with those in the ND group, in liver tissues and eWAT. CIVT remarkably inhibited PPAR-γ and C/EBP-α in liver tissues and eWAT, in a dose-dependent manner. In particular, the CIVT-100 showed significant effects in all factors. These results indicate that CIVT decreased eWAT and liver weight by regulating adipogenesis-related transcription factors.

### 3.6. CIVT Is More Effective Than CIEE, in the Lipid Accumulation

After the mice were sacrificed, the effects of the CI ethanol extract (CIEE) and viscozyme- and tannase-treated ethanol extract (CIVT) on lipid accumulation were compared. As shown in Table 2, CIVT suppressed gain body weight, food efficiency ratio, eWAT and liver weight and orbital fat percentage by 27.26%, 38%, 36.22%, 8.93% and 49.11%, respectively, as compared with levels in the CIEE-100 group. In addition, the serum level of leptin by 25.68% was more decreased and adiponectin by 18.11% was more increased in the CIVT-100 group, than the CIEE-100 group. This result shows that when the enzyme treatment (viscozyme and tannase) is performed, it has a better effect on lipid accumulation.

### 3.7. HPLC Analysis of CIEE and CIVT

The final extraction yields (%) of enzymatic-untreated CI ethanol extract (CIEE) and enzymatic (viscozyme and tannase)-treated ethanol extract CI extract (CIVT) were 27.7% and 48.5%, respectively. The HPLC analysis was performed for the identification of main components in CIEE and CIVT (Figure 5). HPLC showed that luteolin, apigenin, diosmetin and acacetin were main components of CIEE and CIVT. The quantities of the four main components were measured using the calibration curve of each component and compared between CIEE and CIVT. The contents of luteolin, apigenin, diosmetin and acacetin in the CIEE and CIVT are shown in Table 3. As a result, it was confirmed that the contents of luteolin, apigenin, diosmetin and acacetin in CIVT were increased to 4.41 mg/g, 4.04 mg/g, 0.93 mg/g and 2.98 mg/g, respectively.

## 4. Discussion

Worldwide, obesity is rapidly increasing owing to busy lifestyles and irregular eating habits. Obesity is an important health problem because it causes various metabolic diseases [25]. Currently, diet management, exercise and medication are recommended for prevention and treatment of obesity [26]. To treat obesity, anti-obesity medications such as orlistat, sibutramine and rimonabant are commonly used. However, many of these drugs have limits to their use because of severe side effects like anorexia, constipation, insomnia and dizziness [21,27]. Therefore, many researchers are focused on finding substitute therapies with minimal adverse effects, like plant-based herbal or natural products [28].

CI is known as an herb with various health effects, making it potentially useful for treating many diseases. Until now, we have performed many experiments using CI extract to treat obesity [18,19]. As a result, we found that CI was effective in alleviating fat accumulation and inhibiting adipogenesis in HFD-fed obese mice. So, we treated CI extract with the enzymes viscozyme and tannase to make it more effective than untreated CI extract. Subsequently, we investigated the anti-obesity effects of CIVT in HFD-fed obese mice. To study obesity, mice were fed HFD to induce obesity. This is the most common way to mimic obesity in humans and is widely used in obesity research.

In the current study, HFD-fed mice developed obesity and CIVT oral administration alleviated weight gain without affecting food intake levels. However, CIVT reduced food efficiency ratio. Although the amount of food intake was similar, the fact that there was low body-weight gain can be considered advantageous in controlling obesity. This indicates that CIVT oral administration decreased the ability of the mice to convert food intake into body weight (Figure 1). CIVT showed a better effect to suppress weight gain and food efficiency ratio than CIEE at the same dose.

In the CIVT orally administered group, the weight of the organs, including the eWAT, liver, spleen and kidney, was suppressed. Also, by confirming that the weight of liver and eWAT, as well as total weight, decreases, we found that CIVT attenuates fat accumulation in the liver and in the whole body (Table 1). CIVT showed a better effect to inhibit lipid accumulation than CIEE at the same dose.

Lipid metabolism-related biochemical parameters were changed in HFD-fed obesity mice. In contrast, oral administration with CIVT reduced the serum levels of TG, TC and LDL-c and elevated the serum levels of HDL-c, both in a dose-dependent manner (Figure 2A–D). Adipokine, including leptin and adiponectin, is a cytokine that mediates the biological action of the endocrine system [29]. WAT synthesizes and secretes leptin, which plays an important role in controlling food intake and body weight [30] and adiponectin, regulating glucose levels and liver fatty acid oxidation [31]. The leptin concentration increases and the concentration of adiponectin decreases according to the hypertrophy of adipose tissue. According to the present results, in the HFD-fed obese mice, CIVT sharply and dose-dependently suppressed serum levels of leptin and considerably elevated serum levels of adiponectin in a dose-dependent manner (Figure 2E,F). In particular, CIVT-100 showed significant anti-obesity effects on all lipid metabolism-related factors.

We measured total lipid in the liver and found that CIVT dose-dependently reduced the total lipid quantity in HFD-fed mice (Figure 3E).

The effect of CIVT on visceral fat accumulation was investigated using micro-CT (Figure 3A,B). In addition, H&E staining was performed to observe the histological alteration and hypertrophy of adipose tissue (Figure 3C,D). As a result, CIVT was observed to reduce the fat volume in ventral and inhibit adipocyte hypertrophy in adipose tissue.

The adipogenic differentiation of pre-adipocytes into mature adipocytes is manifested by changes in gene expression associated with adipogenesis [15]. Transcription factors such as PPARs and C/EBP-α are known to play a crucial role in adipogenesis [32]. PPARs are important nuclear receptor transcription factors that regulate the proliferation and differentiation of pre-adipocytes. They control the expression of genes involved in fatty acid oxidation and synthesis and adipogenesis [33]. PPARs, including PPAR-α and PPAR-γ, are expressed in the adipose tissue and liver. PPAR-α is known to play a role in controlling nutrient metabolism, like gluconeogenesis and amino acid metabolism [15]. In addition, the PPAR-γ is highly expressed in adipose tissue and known to regulate adiponectin, an adipose-derived hormone that enhances insulin sensitivity in the liver [34]. When C/EBP-α is activated, insulin resistance is reduced and adipogenic differentiation is induced, converting pre-adipocytes into mature adipocytes and promoting the development of adipose tissues [11]. We found that the protein expression levels of PPAR-α decreased and those of PPAR-γ and C/EBP-α increased, in mice fed HFD. However, upon oral administration with different concentrations of CIVT, PPAR-α level was increased and PPAR-γ and C/EBP-α levels were considerably decreased in the liver tissue and adipose tissue (Figure 4). These results indicate that CIVT effectively alleviate adipogenesis by regulating the adipogenic genes. This finding is similar to that in our previous study, which demonstrated that CIEE has anti-obesity effects in HFD-fed mice. [19].

To compare the anti-obesity effect between untreated CI ethanol extract (CIEE) and enzyme-treated CI ethanol extract (CIVT), we examined the difference in effect of the same amount of the two substances on adipogenesis-related gene expression. As a result, CIVT decreased gain body weight, food efficiency ratio, eWAT and liver weight, orbital fat percentage and leptin levels compared with CIEE. And CIVT increased adiponectin levels compared with CIEE (Table 2).

As components of the CI, flavonoids, terpenoids and phenolic compounds have been reported [35]. Flavonoids in the extracts of CIEE are also mainly existed in the form of glycosides. These glycosides have low absorption ability in the body and are disadvantageous in that they must be hydrolysed by various enzymes in order to increase the absorption ability. The aglycone produced by hydrolysis is an efficient form that can be absorbed immediately and is useful for digestion and absorption [36]. In several studies, enzymatic hydrolysates of plants have been reported to have many biological properties such as antioxidants, anticoagulants and antiproliferative activity [37].

The major flavonoids of the CI extract are reported as luteolin, apigenin, diosmetin and acacetin [38]. Luteolin, apigenin and acacetin are known to improve obesity by inhibiting adipogenesis and lipid absorption [39,40,41]. By treating the CI extract with enzymes, we converted the flavonoid glycosides to aglycones, increased bioavailability. Upon comparing the amounts of CIEE and CIVT aglycones components using HPLC, it was confirmed that CIVT contained compounds 2.81-fold (luteolin), 2.68-fold (apigenin), 3.88-fold (diosmetin) and 3.73-fold (acacetin) higher than CIEE (Figure 5, Table 3). These results suggest that the enzymatic treatment of CI extracts leads to better anti-obesity effect by increasing the contents of anti-obesity components.

In conclusion, our results showed that CIVT administration has an effective anti-obesity effect in HFD-fed mice. Also, anti-obesity effect of CIVT was considerably enhanced when compared to that of CIEE. The reason for the superior effect of CIVT is thought to be the change in the contents of flavonoids, which are inhibiting adipogenesis, upon the treatment with the enzymes. It supports that the enzymatic treatment with viscozyme and tannase remarkably enhances anti-obesity efficacy and bioavailability of CI and that CIVT could serve as potential candidate for the prevention or treatment of obesity.

## Figures and Tables

**Figure 1 nutrients-11-00269-f001:**
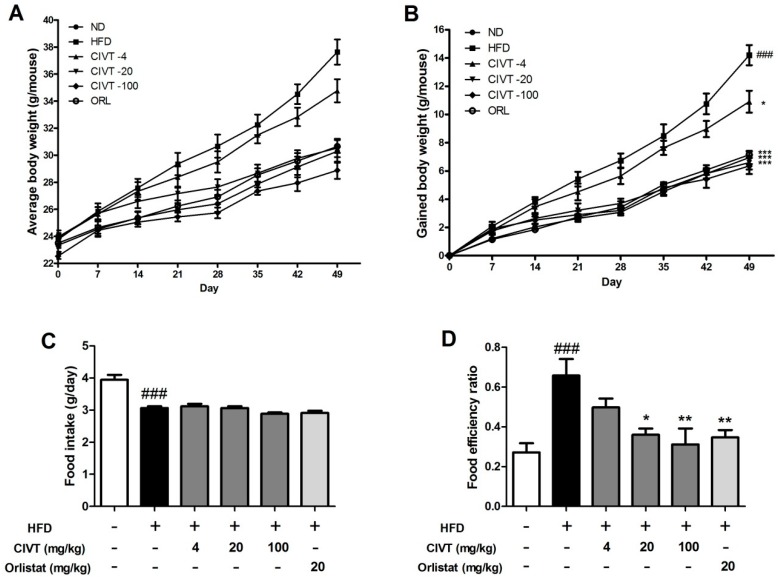
Effects of CIVT on body weight gain, food intake and food efficiency ratio in HFD-fed obese mice. The mice (*n* = 6) were fed normal chow diet (ND) or high-fat diet (HFD). HFD fed mice were administered CIVT (4, 20 or 100 mg/kg) or orlistat (20 mg/kg) orally once a day. During the 7-week experimental period, average body weight (**A**), gained body weight (**B**), food intake (g per day) (**C**) and food efficiency ratio (**D**) was measured once a week. Food efficiency ratio = Gained body weight (g)/Food intake during trial period (g). Values represent the means ± SEMs (*n* = 6). Data were analysed by Bonferroni correction for *post-hoc* test. *^###^ p* < 0.001 versus the ND group; ** p* < 0.05, *** p* < 0.01 and **** p* < 0.001 versus the HFD group.

**Figure 2 nutrients-11-00269-f002:**
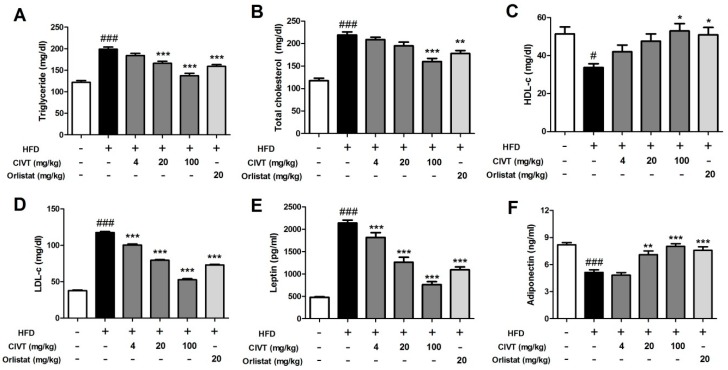
Effects of CIVT on serum levels of biochemical parameters in HFD-fed obese mice. After the 7-week experimental period, serum levels of Triglyceride (**A**), Total cholesterol (**B**), HDL-c (**C**) and LDL-c (**D**) were measured using commercial kits. Serum levels of Leptin (**E**) and Adiponectin (**F**) were measured using ELISA. Values represent the means ± SEMs (*n* = 6). Data were analysed by Bonferroni correction for *post-hoc* test. *^###^ p* < 0.001 versus the ND group; ** p* < 0.05, *** p* < 0.01 and **** p* < 0.001 versus the HFD group.

**Figure 3 nutrients-11-00269-f003:**
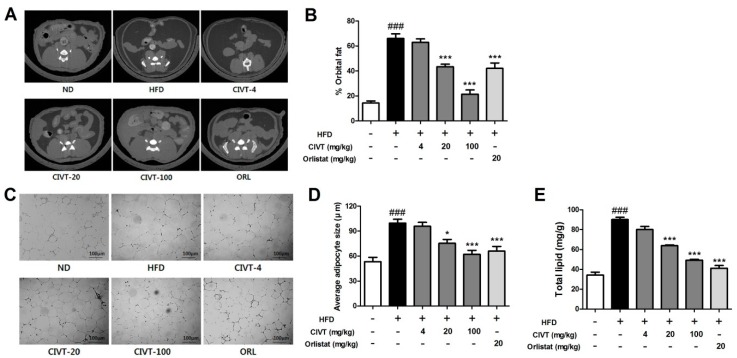
Effects of CIVT on fat deposition and histological alterations and adipocytes size of eWAT and total lipid levels in the liver tissues in HFD-fed obese mice. Micro-CT images were obtained using Skyscan1076 micro-CT scanner. Representative micro-CT images in each group (*n* = 6) were selected (**A**). Percentage (%) of orbital fat was analysed using Skyscan program (**B**). Epididymal WAT (eWAT) was removed from the mice after they were sacrificed. The histological alterations of the tissue were observed using H&E staining (**C**). Scale bar: 100 μm, Original magnification: 100×. Size of adipocytes from eWAT was measured using Photoshop program (**D**). Total lipids were analysed (**E**). Values represent the means ± SEMs (*n* = 6). Data were analysed by Bonferroni correction for *post-hoc* test. *^###^ p* < 0.001 versus the ND group; ** p* < 0.05 and **** p* < 0.001 versus the HFD group.

**Figure 4 nutrients-11-00269-f004:**
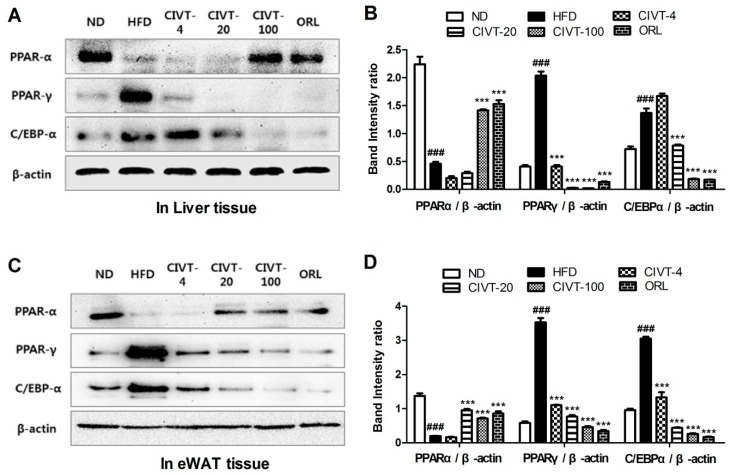
Effects of CIVT on the protein expression levels of transcription factors of adipogenesis in HFD-fed obese mice. Protein expression levels of PPAR-α, PPAR-γ and C/EBP-α in liver tissues were analysed using western blotting (**A**). The intensities of the relative band in the western blot are shown as bar graph as follow: PPAR-α/ β-actin, PPAR-γ/ β-actin and C/EBP-α/ β-actin (**B**). Protein expression levels of PPAR-α, PPAR-γ and C/EBP-α in eWAT tissues were analysed using western blot (**C**). The intensities of the relative band of western blot were shown as a bar graph as follow: PPAR-α/ β-actin, PPAR-γ/ β-actin and C/EBP-α/ β-actin (**D**). Values represent the means ± SEMs of three independent experiments. Data were analysed by Bonferroni correction for *post-hoc* test. *^###^ p* < 0.001 versus the ND group; **** p* < 0.001 versus the HFD group.

**Figure 5 nutrients-11-00269-f005:**
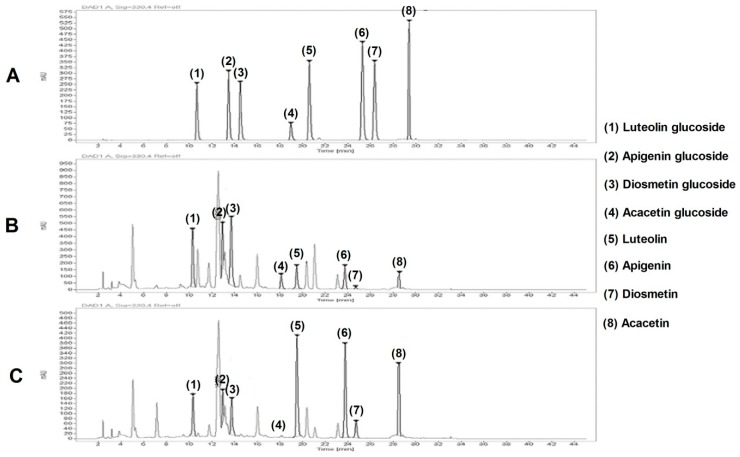
HPLC analysis of CIEE and CIVT. HPLC chromatogram of the standard mixtures (**A**), CIEE (**B**) and CIVT (**C**) was obtained according to the Materials and Methods.

**Table 1 nutrients-11-00269-t001:** Effects of CIVT on organs weight in HFD-fed obesity mice.

	ND	HFD	CIVT-4	CIVT-20	CIVT-100	ORL
eWAT (g)	0.56 ± 0.14	2.03 ± 0.08 ^#^	1.83 ± 0.12	0.87 ± 0.17 ^c^	0.81 ± 0.09 ^c^	1.20 ± 0.14 ^b^
Liver (g)	1.11 ± 0.06	1.55 ± 0.08 ^#^	1.40 ± 0.06	1.23 ± 0.07 ^a^	1.02 ± 0.03 ^c^	1.31 ± 0.04
Spleen (g)	0.09 ± 0.01	0.12 ± 0.01 ^#^	0.11 ± 0.01	0.09 ± 0.01 ^a^	0.09 ± 0.01 ^b^	0.09 ± 0.01 ^b^
Kidney (g)	0.33 ± 0.01	0.42 ± 0.01	0.39 ± 0.02	0.37 ± 0.01	0.36 ± 0.01	0.38 ± 0.01

Values represent the means ± SEMs (*n* = 6). Data were analysed by Bonferroni correction for *post-hoc* test. ^#^
*p* < 0.001 versus the ND group; ^a^
*p* < 0.05, ^b^
*p* < 0.01 and ^c^
*p* < 0.001 versus the HFD group.

**Table 2 nutrients-11-00269-t002:** Comparison of anti-obesity effects of CIEE and CIVT.

	ND	HFD	CIEE-100	CIVT-100
Gained body weight (g/mouse)	6.95 ± 0.46	14.20 ± 0.72 ^#^	8.73 ± 0.31 ^c^	6.35 ± 0.57 ^c, d^
Food efficiency ratio	0.27 ± 0.05	0.65 ± 0.08 ^#^	0.5 ± 0.03 ^a^	0.31 ± 0.08 ^b^
eWAT (g)	0.56 ± 0.14	2.03 ± 0.08 ^#^	1.27 ± 0.12 ^c^	0.81 ± 0.09 ^c, d^
Liver (g)	1.11 ± 0.06	1.55 ± 0.08 ^#^	1.12 ± 0.05 ^a^	1.02 ± 0.03 ^c, d^
Orbital fat (%)	14.32 ± 1.69	66.06 ± 3.65 ^#^	42.1 ± 4.31 ^c^	21.45 ± 3.38 ^c, e^
Leptin (pg/mL)	477.0 ± 11.37	2140.0 ± 65 ^#^	1025 ± 112 ^c^	761.8 ± 63.7 ^c, e^
Adiponectin (ng/mL)	8.18 ± 0.25	5.14 ± 0.29 ^#^	6.79 ± 0.20 ^c^	8.02 ± 0.29 ^c, d^

Values represent the means ± SEMs (*n* = 6). Data were analysed by Bonferroni correction for *post-hoc* test. ^#^
*p* < 0.001 versus the ND group; ^a^
*p* < 0.05, ^b^
*p* < 0.01 and ^c^
*p* < 0.001 versus the HFD group. ^d^
*p* < 0.05 and ^e^
*p* < 0.01 versus the CIEE group.

**Table 3 nutrients-11-00269-t003:** Comparison of the main components from CIEE and CIVT.

	CIEE	CIVT
Yields (%)	27.7	48.5
Luteolin glucoside (mg/g)	13.89	6.86
Apigenin glucoside (mg/g)	13.00	5.07
Diosmetin glucoside (mg/g)	17.80	6.32
Acacetin glucoside (mg/g)	2.23	0.71
Luteolin (mg/g)	1.57	4.41
Apigenin (mg/g)	1.51	4.04
Diosmetin (mg/g)	0.24	0.93
Acacetin (mg/g)	0.80	2.98

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
