# Peer review of "Effect of Enzymatic Treatment of Chrysanthemum Indicum Linné Extracts on Lipid Accumulation and Adipogenesis in High-Fat-Diet-Induced Obese Male Mice"

_nutrients, 2019, doi:10.3390/nu11020269_

Reviewer 1 Report

1. Title

a. The title should include exactly the population studied, thus indicating that it is male mice (2-4)

 2. Abstract

a. It would be interesting to include the total number of mice included and their sex (23)

b. Please include the explanation of the initials eWAT (27)

 3. Materials and methods

a. It would be interesting to know why epididymal white adipose tissue is used for histological analysis and not other tissues (158)

4. Discussion

a. It would be nice to include a current quote about the incidence of obesity and the public health problem that it involves (314-5)

b. The appointment 24 is little current to start the sentence with a "currently", it would be nice to update this appointment to include the current indications of the treatment of obesity (315-7)

 7. References

to. The bibliography is extensive and relatively updated with 17.5% of the citations of the last five years. It would be interesting to include the quotes mentioned above in relation to obesity

Author Response

Reviewer 1.

Comments and Suggestions for Authors

 1. Title

a. The title should include exactly the population studied, thus indicating that it is male mice (2-4).

Answer:

Thank you for your kind checking. As your comments, we added the word ‘Male’ in the title.

 2. Abstract

a. It would be interesting to include the total number of mice included and their sex (23).

Answer:

Thank you for your kind opinion. As your comments, we added the word ‘male’ in the abstract (line 24).

 b. Please include the explanation of the initials eWAT (27).

Answer:

Thank you for your point out. We had a mistake in the abstract. As your indication, we added the full name of initials ‘eWAT’ (line 29 : epididymal white adipose tissue (eWAT).

 3. Materials and methods

a. It would be interesting to know why epididymal white adipose tissue is used for histological analysis and not other tissues (158).

Answer:

Thank you for your kind advice. At the sacrifice of obesity-induced mice, we can observe a marked increase in epididymal white adipose tissue in the abdomen. Therefore, most of the researchers studying obese mice are using the epididymal white adipose tissue to analyze. So, we also analyzed representatively epididymal white adipose tissue among all the adipose tissues of mice.

We have added a description in the manuscript (line 163-164).

“It is observed that the eWAT is markedly increased in abdominal cavity of obese-induced mice. Therefore, white adipose tissues of all mice were analyzed by observing representatively eWAT.”

 4. Discussion

a. It would be nice to include a current quote about the incidence of obesity and the public health problem that it involves (314-5).

Answer:

Thank you for your good advice. Based on your advice, we included the reference [25] (line 323).

 b. The appointment 24 is little current to start the sentence with a "currently", it would be nice to update this appointment to include the current indications of the treatment of obesity (315-7).

Answer:

Thank you for your keen point out. As your comments, we changed reference [26] (line 323-325).

(Due to the addition of reference [24] according to review 4-a, the existing [24] has been changed to [26].)

 7. References

to. The bibliography is extensive and relatively updated with 17.5% of the citations of the last five years. It would be interesting to include the quotes mentioned above in relation to obesity.

Answer:

Thank you for your advice. Based on your feedback, we reviewed the reference again and updated several references.

Reviewer 2 Report

In their manuscript, Lee et al. describe their study on the obesity-inhibitory effect of Chrysanthemum indicum (CI) in enzyme treated and untreated forms to high fat diet (HFD) fed mice. The study is interesting and shows some promising results; however, some work is needed for clarity and controls.

1)   The entire paper could use some minor editing for English language to elevate readability and understandability.

2)   Line 45 (and others): it is unclear what the authors mean by an “effective” ingredient as it is not specified what effectiveness is being measured. Perhaps it would be better to say “bioactive” or another way to describe these components of the plant matter.

3)   Unfortunately the animal experiments described are highly unclear. It seems the experiments are missing vital controls (normal chow with all levels of CIVT treatment and CIEE treatment at 20 mg/kg). Can the authors please clarify the dietary treatments and include in any necessary controls? In addition, a figure or outline of treatments would be ideal.

4)   Similar to my previous comment, it is unclear on the exact experimental design in terms of numbers of mice and how many times experiments were repeated. The authors state they used n=6 mice/group; however, this seems highly underpowered for an adequate study. How many times was each set of experiments repeated? Please also include any and all power calculations that were performed before the experimentation was conducted to show that adequate power was considered.

5)   Were mice housed individually or in groups? Were they pair fed or just fed ad libitum with amount of food intake measured? This kind of descriptive information is vital for any feeding study.

6)   Along with previous comments, was a vehicle control used for all groups? If not, this is a necessary control.

7)   The authors give no explanation for the orlistat treatment. This needs to be fully explained and controlled.

8)   Line 194: Please define ORL in the first instance of use of the abbreviation.

9)   Line 374: I do not think it is accurate to say that CIVT can “inhibit adipogenesis” as this has not been directly shown and the exact mechanisms of the treatment have not been investigated. It is very important to do further work to determine the specific mechanism behind the efficacy of this treatment on the prevention of obesity.

10)                   Line 377-378: It does not appear that side effects have been measured in this study and I find it doubtful that any meaningful side effects can be measured without doing human trails. Please consider removing this statement.

11)                   Line 403: should be “adipogenesis”?

12)                   It would be ideal for the authors to show that this treatment reduces obesity in mice already fed a HFD versus those that are given the drug with the diet. Reduction of existing obesity is equally, if not more, important than preventing obesity development.

Author Response

Reviewer 2.

Comments and Suggestions for Authors

 In their manuscript, Lee et al. describe their study on the obesity-inhibitory effect of Chrysanthemum indicum (CI) in enzyme treated and untreated forms to high fat diet (HFD) fed mice. The study is interesting and shows some promising results; however, some work is needed for clarity and controls.

 1) The entire paper could use some minor editing for English language to elevate readability and understandability.

Answer:

Thank you for your interest and advice in our manuscript. We have already received an English language editing. After some changed according to reviewers comments, we have reviewed it once more to elevate readability and understandability.

 2) Line 45 (and others): it is unclear what the authors mean by an “effective” ingredient as it is not specified what effectiveness is being measured. Perhaps it would be better to say “bioactive” or another way to describe these components of the plant matter.

Answer:

Thank you for your opinion. As your comments, we changed the word “effective” to “bioactive” (line 46).

 3) Unfortunately the animal experiments described are highly unclear. It seems the experiments are missing vital controls (normal chow with all levels of CIVT treatment and CIEE treatment at 20 mg/kg). Can the authors please clarify the dietary treatments and include in any necessary controls? In addition, a figure or outline of treatments would be ideal.

Answer:

We have already confirmed the effect of CIEE on obesity-induced mice in previous article (Reference [18]). So, it was thought that the CIEE experiment again with these concentrations (4, and 20 mg/kg) was overlapped and meaningless. Therefore, only the most effective concentrations of CIEE and CIVT 100 mg/kg were tested for comparison purpose.

 4) Similar to my previous comment, it is unclear on the exact experimental design in terms of numbers of mice and how many times experiments were repeated. The authors state they used n=6 mice/group; however, this seems highly underpowered for an adequate study. How many times was each set of experiments repeated? Please also include any and all power calculations that were performed before the experimentation was conducted to show that adequate power was considered.

Answer:

Thank you. The minimum animal population for experiment statistics is three. In many studies, more than 5 mice per group are being used within the experimental group for uniformity. Thus, the number of mice per group in our study was never small, and the experiment was conducted within the range of statistical significance. Each experiment was repeated three times. We've added this content to each figure legend.

 5) Were mice housed individually or in groups? Were they pair fed or just fed ad libitum with amount of food intake measured? This kind of descriptive information is vital for any feeding study.

Answer:

Thank you for your good advice. The mice were housed in groups and fed ad libitum. To measure food intake, we measured every week the weight of food that the mice consumed for a week. Depending on your opinion, these sentences have been added to the 2.4. Animals and Treatment section of the Material and Method section (line 127-128).

 6) Along with previous comments, was a vehicle control used for all groups? If not, this is a necessary control.

Answer:

Thank you. All groups were vehicle controlled.

 7) The authors give no explanation for the orlistat treatment. This needs to be fully explained and controlled.

Answer:

Thank you for your opinion. We have already explained briefly about 'orlistat treatment' in the Material and Method section and mentioned in the Discussion section line 325-329. However, we thought that our explanation was not enough according to your opinion. So, we added additional clarification and reference (line 123-124, Reference [20]).

 8) Line 194: Please define ORL in the first instance of use of the abbreviation.

Answer:

Thank you for your point out. As your comments, we added the definition of ORL (line 200).

 9) Line 374: I do not think it is accurate to say that CIVT can “inhibit adipogenesis” as this has not been directly shown and the exact mechanisms of the treatment have not been investigated. It is very important to do further work to determine the specific mechanism behind the efficacy of this treatment on the prevention of obesity.

Answer:

Thank you for your kind opinion. We wrote these sentences “inhibit adipogenesis” due to observe that CIVT significantly modulates the expression of adipogenesis-related gene expression. However, as your opinion, we have changed the word 'inhibit' to 'alleviate' because the specific mechanism of CIVT has not been investigated. (line 383). The detailed and specific mechanism of CIVT's prevention and inhibition of obesity will be studied in a follow-up study.

 10) Line 377-378: It does not appear that side effects have been measured in this study and I find it doubtful that any meaningful side effects can be measured without doing human trails. Please consider removing this statement.

Answer:

Thank you for your kindly advice. As your advice, we removed the word ‘few side effects’.

 11) Line 403: should be “adipogenesis”?

Answer:

Thank you. We found mistake in your keen point. As your comment, we changed the word (line 411).

 12) It would be ideal for the authors to show that this treatment reduces obesity in mice already fed a HFD versus those that are given the drug with the diet. Reduction of existing obesity is equally, if not more, important than preventing obesity development.

Answer:

Thank you very much for your good feedback. Based on your opinion, in a follow-up study, we will confirm the reduction in obesity by administering the drug after feeding HFD.

Round  2

Reviewer 2 Report

The authors have addressed the majority of my previous concerns to the best of their ability. Thank you.

Author Response

Thank you for the reviews.